# Morroniside Inhibits Inflammatory Bone Loss through the TRAF6-Mediated NF-κB/MAPK Signalling Pathway

**DOI:** 10.3390/ph16101438

**Published:** 2023-10-10

**Authors:** Jirimutu Xiao, Qiuge Han, Ziceng Yu, Mengmin Liu, Jie Sun, Mao Wu, Heng Yin, Jingyue Fu, Yang Guo, Lining Wang, Yong Ma

**Affiliations:** 1Laboratory of New Techniques of Restoration & Reconstruction, Institute of Traumatology & Orthopedics, Nanjing University of Chinese Medicine, Nanjing 210023, China; 15034995889@163.com (J.X.); hqg@njucm.edu.cn (Q.H.); 20200502@njucm.edu.cn (Z.Y.); 829004@njucm.edu.cn (M.L.); 20193017@njucm.edu.cn (J.S.); 20220013@njucm.edu.cn (J.F.); drguoyang@njucm.edu.cn (Y.G.); 2School of Mongolia Medicine, Inner Mongolia Medical University, Hohhot 010110, China; 3School of Chinese Medicine · School of Integrated Chinese and Western Medicine, Nanjing University of Chinese Medicine, Nanjing 210023, China; 4Jiangsu CM Clinical Innovation Center of Degenerative Bone & Joint Disease, Wuxi TCM Hospital Affiliated to Nanjing University of Chinese Medicine, Wuxi 214071, China; wxkfhmm@sina.cn (M.W.); wxzy011@njucm.edu.cn (H.Y.)

**Keywords:** morroniside, osteoporosis, inflammation, osteogenic differentiation, TRAF6-mediated MAPK/NF-κB signaling pathway

## Abstract

Osteoporosis is a chronic inflammatory disease that severely affects quality of life. *Cornus officinalis* is a Chinese herbal medicine with various bioactive ingredients, among which morroniside is its signature ingredient. Although anti-bone resorption drugs are the main treatment for bone loss, promoting bone anabolism is more suitable for increasing bone mass. Therefore, identifying changes in bone formation induced by morroniside may be conducive to developing effective intervention methods. In this study, morroniside was found to promote the osteogenic differentiation of bone marrow stem cells (BMSCs) and inhibit inflammation-induced bone loss in an in vivo mouse model of inflammatory bone loss. Morroniside enhanced bone density and bone microstructure, and inhibited the expression of IL6, IL1β, and ALP in serum (*p* < 0.05). Furthermore, in in vitro experiments, BMSCs exposed to 0–256 μM morroniside did not show cytotoxicity. Morroniside inhibited the expression of IL6 and IL1β and promoted the expression of the osteogenic transcription factors Runx2 and OCN. Furthermore, morroniside promoted osteocalcin and Runx2 expression and inhibited TRAF6-mediated NF-κB and MAPK signaling, as well as osteoblast growth and NF-κB nuclear transposition. Thus, morroniside promoted osteogenic differentiation of BMSCs, slowed the occurrence of the inflammatory response, and inhibited bone loss in mice with inflammatory bone loss.

## 1. Introduction

Osteoporosis is a systemic metabolic disease caused by various factors that change bone microstructure and increase bone fragility, thereby seriously reducing quality of life. The main risk factors for osteoporosis are age, low body mass index, early menopause, and Chinese ethnicity, together with secondary factors. The decline in ovarian function in women after menopause leads to a decrease in estrogen levels and changes in hormone levels (sensitivity), such as calcitonin and parathyroid hormones. In patients with senile osteoporosis, bone marrow-derived mesenchymal stem cells age, their regenerative ability is weakened, their growth cycle is stagnated, and their level of osteogenic differentiation is reduced, resulting in changes in the bone microstructure and aging of extramedullary tissues [1,2,3]. Due to an increasingly large elderly population, the number of people with osteoporosis has increased to approximately 200 million globally [4]. Recent studies have shown that chronic inflammation, such as ulcerative colitis and systemic lupus erythematosus, affects bone metabolism and leads to bone loss [5,6]. The differentiation of bone marrow stem cells (BMSCs) at the appropriate times and locations is critical for bone formation. Inflammatory responses are critical for the initiation of bone repair [7]. When inflammation occurs, inflammatory cells recruit BMSCs, which direct their migration and differentiation, and the bone marrow microenvironment changes considerably [7], promoting TNF-receptor-related factor 6 (TRAF6) hemagglutination and autoubiquitination [8,9], which provide signals that inhibit osteogenesis through mediating various signaling molecules and activating the NF-κB and MAPK signaling pathways [10,11], thus providing a signal to inhibit osteogenesis [12]. Therefore, the identification of antagonistic signs of osteogenic differentiation of MSCs from the bone marrow microenvironment is key to elucidating the molecular mechanisms of inflammatory bone loss [13,14,15,16].

Bisphosphonates and dezumumab are currently used to treat osteoporosis and exert a strong inhibitory effect on bone resorption by reducing the formation of osteoclasts. Their side effects include gastrointestinal complications, musculoskeletal pain, jaw necrosis, and renal toxicity. Atypical fragility fractures are caused by bisphosphonates [17,18]. Dezumumab (DMAB) inhibits the interaction of RANKL and RANK with inactivate osteoclasts, thereby reducing bone resorption, increasing bone mass, and improving bone strength. However, the effects of DMAB treatment are reversible, and bone turnover tends to increase rapidly after discontinuation, followed by bone loss. An increased risk of fracture has been reported during this phase of high bone turnover, and its side effects are severe [19,20,21]. In this context, many researchers have focused on the safety and effectiveness of traditional Chinese medicine (TCM), among which numerous methods are presently used to treat osteoporosis.

The main active ingredient in *Cornus officinalis* is morroniside. Morroniside, a small-molecule monomer compound, has been shown to have many biological effects, such as anti-inflammatory [22], anti-apoptotic [23], and anti-oxidative stress [24] effects. Morroniside promotes osteoblast differentiation [25], attenuates BMSC dysfunction mediated by high glucose levels [26], and promotes BMSC proliferation [27]. Moreover, morroniside inhibits the expression of inflammatory cytokines and slows the onset of inflammation [28,29]. The inflammatory mediators of osteogenic differentiation in BMSCs are rarely reported to be affected by morroniside.

Importantly, the specific anti-osteoporotic effects of morroniside on lipopolysaccharide (LPS)-induced osteogenic differentiation dysfunction and inflammatory bone loss in mouse models of BMSCs have not yet been reported. Based on this, this study utilzed isolated mice BMSCs in vitro, stimulated by LPS, to establish an inflammatory model of BMSCs, and established a mice model of inflammatory bone loss in vivo to explore the effect of morroniside on inflammatory bone loss, aiming to provide a new concept for the prevention and treatment of osteoporosis. Moreover, to some extent, this study hopes to increase the development and utilization of cornus officinalis.

## 2. Results

### 2.1. Morroniside Increases Bone Mineral Density and Improves Bone Microstructure in Lipopolysaccharide-Induced Inflammatory Bone Loss Mice

In mice with LPS-induced inflammatory bone loss, the results of micro-CT and HE staining showed that the trabecular shape of the femur in the control group was complete, tightly arranged, and orderly; the intertrabecular connections were reticular; the bone marrow cavity was small; and the number of bone marrow cells was abundant. In the LPS group, the trabecular bone was sparse, the number of bone marrow cells was significantly reduced, the cell arrangement was disordered, the bone marrow cavity was enlarged, and a part of it was hollow, indicating typical osteoporosis. Three-dimensional (3D) BMD, BV/TV, and Tb.Th significantly decreased, while Tb.Sp significantly increased. After the administration of morroniside, the trabecular bone arrangement became more regular; the number of trabecular bone cells, as well as the number of cells in the bone marrow cavity, 3D BMD, BV/TV, and Tb.Th, increased; and the bone marrow cavity and Tb.Sp decreased (Figure 1A–F).

The results of biomechanical tests indicated that the maximum load, maximum deflection, and stiffness of the mice in the LPS group were significantly inhibited, with both morroniside and alendronate sodium increasing these three parameters to varying degrees. Treatment with alendronate sodium and morroniside had the same effect on increasing the maximum deflection and stiffness, although alendronate was slightly more effective than morroniside in increasing the maximum load (Figure 1G–I).

### 2.2. Morroniside Inhibits Inflammatory Bone Loss in Mice through the TRAF6-Mediated NF-κB/MAPK Signalling Pathway

The expression of IL-6, IL-1β, ALP, and TRAP in the sera of mice in the LPS group was significantly increased. The morroniside and alendronate treatments inhibited the expression of IL-6, IL-1β, and ALP in the sera of the mice, producing the same inhibitory effects. Alendronate sodium treatment inhibited TRAP expression more effectively than morroniside (Figure 1J−M). Osteocalcin (OCN) and Runx2 expression in the tibia was detected using qPCR and Western blotting, respectively. The results showed that morroniside and alendronate sodium treatment both promoted the expression of OCN and Runx2, while the effect of alendronate sodium was stronger than that of morroniside (Figure 2D,E,J,L,M).

While the mRNA expression of TRAF6 increased in the LPS group, the morroniside and alendronate sodium treatments reduced the mRNA expression of TRAF6, with both treatments having similar inhibitory effects (Figure 2C). The expression of TRAF6 in the femur tissue was significantly increased in the LPS group, while morroniside treatment significantly inhibited the expression of TRAF6, with a slightly weaker inhibitory effect than that of alendronate sodium (Figure 2A,B). Using Western blotting to detect the protein expression levels of TRAF6, p-p65, p-Erk, and p-p38, the levels were found to be higher in the LPS group. Both morroniside and alendronate sodium inhibited the expression of TRAF6, p-p65, p-ERK, and p-p38; however, alendronate sodium treatment had a stronger inhibitory effect (Figure 2F−I,K).

### 2.3. Identification of BMSCs and Effect of Morroniside on Cell Activity

The positivity rates for CD90, CD29, CD34, and CD45 were found to be 98.6%, 99.6%, 3.1%, and 5.29%, respectively (Figure 3C–F).

To evaluate the cytotoxicity of morroniside, BMSCs were cultured with different concentrations of morroniside for 48 h. As shown in Figure 3, when 512 μM morroniside was applied for 48 h, cytotoxicity to BMSCs was observed; by contrast, the other concentrations of morroniside showed no cytotoxicity to BMSCs; however, this difference was not statistically significant compared with the group without morroniside treatment (Figure 3B). Under a microscope, the cells showed spindle-shaped and polygonal adherent growth with a uniform morphology, consistent with the characteristics of BMSCs (Figure 3G–J).

### 2.4. Morroniside Increases LPS-Induced Osteogenic Differentiation In Vitro

The ELISA results showed that the expression of IL-1β and TNF-α was increased in the supernatant of BMSCs stimulated with LPS. Morroniside reduced the expression of IL-1β and TNF-α (Figure 3K,L). The osteogenic effect of the BMSCs was detected using ALP and alizarin red staining. The number of positive cells was significantly lower in the LPS group than in the control group. The number of positive cells was significantly higher in the low- and high-dose morroniside groups (Figure 3M,N). The in vitro expression of Runx2 and OCN was detected through Western blotting, and the results were consistent with those of ALP and alizarin red staining. The expression of Runx2 and OCN was lower in the LPS group than in the control group. Compared to the LPS group, the protein expression levels of Runx2 and OCN were significantly upregulated in the low- and high-dose morroniside groups (Figure 4A–C).

### 2.5. Morroniside Inhibits the LPS-Induced Activation of TRAF6-Mediated NF-κB/MAPK Signaling Pathway in BMSCs

The results showed that TRAF6 and p-p65/p65 protein expression was upregulated in the LPS group. However, morroniside treatment reduced TRAF6 and p-p65/p65 protein expression. Furthermore, morroniside inhibited the LPS-induced reduction in BMSC differentiation, which is also related to the MAPK signaling pathway. Compared to the control group, the protein levels of p-ERK/ERK and p-p38/p38 were higher in the LPS group. Compared to the LPS group, the protein expression levels of p-Erk/Erk and p-p38/p38 were lower in the morroniside group (Figure 4D–H). Immunofluorescence assays showed that morroniside inhibited the expression of p65 (Figure 4I).

## 3. Discussion

Morroniside is an iridoid glycoside and has attracted increasing attention in recent years in terms of its medicinal value. Studies have shown that morroniside can regulate protein expression to improve inflammatory skeletal muscle atrophy [30] and can reduce cartilage destruction by reducing the expression of inflammatory mediators (such as Cox-2, Mmp3, and Mmp13) in articular cartilage [28]. More importantly, it can promote the differentiation of osteoblasts, inhibit the differentiation of osteoclasts, and enhance bone mineral density, to play an anti-osteoporosis role [25], which also provides a certain basis for our study on the anti-inflammatory bone loss of mice using morroniside. Inflammation is closely related to osteoporosis. In this study, we established a model of LPS-induced inflammatory bone loss in BMSCs to determine whether morroniside promotes the osteogenic differentiation of BMSCs. Morroniside treatment was found to inhibit the activation of the NF-κB and MAPK signaling pathways and p65 nuclear translocation by TRAF6, as well as to promote the osteogenic differentiation of BMSCs that had been induced by LPS, which inhibited inflammatory bone loss.

Inflammation is an important factor involved in bone loss. The relationship between TRAF6, NF-κB, MAPK, and osteoporosis, which are related to pathogenesis, is becoming increasingly evident [31,32], further demonstrating the role of inflammation in the induction of osteoporosis and suggesting the possibility of treating osteoporosis through inhibiting inflammation. Morroniside increases the expression of Runx2, promotes osteoblast differentiation, and inhibits the osteoporosis caused by ovariectomies [25]. The NF-κB and MAPK signaling pathways can also be inhibited by morroniside, preventing inflammation and oxidative damage [33,34].

Runx2 is a unique transcriptional regulator of osteoblast differentiation that controls the expression of osteogenic genes. The target genes involved in the Runx2 interactions were OCN and osteocalcin-specific cis-acting elements (OSE). OCN promotes osteoblast formation [11,35,36]. By establishing a mouse model of inflammatory bone loss, HE and Micro-CT methods were used to determine that the mice treated with LPS showed typical osteoporosis. The biomechanical parameter analysis results showed that there was no significant difference in maximum load between the model group and the control group, which may have been due to the relatively small number of samples, and the number of samples could be further increased. Furthermore, it was found through ELISA that morroniside could inhibit the expression of IL6, IL1β, ALP, and TRAP in mice serum. Next, we detected the expression of the osteogenic genes Runx2 and OCN. The results showed that morroniside could promote the expression of Runx2 and OCN mRNA and protein, indicating that morroniside could slow down the occurrence of inflammation, regulate bone metabolism, and promote osteogenic differentiation. The expression of related proteins in the TRAF6-mediated NF-κB/MAPK signaling pathway was detected using Western blot. The results showed that morroniside could inhibit the expression of TRAF6, p-p65, p-Erk, and p-p38. It is suggested that morroniside may slow down the inflammatory response through the TRAF6-mediated NF-κB/MAPK signaling pathway, promote osteogenic differentiation, and inhibit inflammatory bone loss. Therefore, whether morroniside can play a role in promoting osteogenic differentiation of BMSCs remains to be further studied.

A close relationship exists between inflammation and the osteogenic differentiation of mesenchymal stem cells in bone marrow. In inflammatory microenvironments, genes undergo genome-wide changes in expression [37,38], resulting in the secretion of signaling molecules and the regulation of innate immune responses [39,40], contributing to tissue repair and affecting a wide range of biological processes, including proliferation, differentiation, migration, and apoptosis. The TRAF6-mediated NF-κB and MAPK signaling pathways are also involved in inflammatory responses [41] and are essential for bone formation and regeneration [42]. TRAF6 can activate the NF-κB and MAPK signaling pathways, inhibit the expression of IκBα, and promote the expression of p65 [43]. Research shows that morroniside exerts LPS-induced anti-inflammatory and antioxidant effects by targeting the TLR4/NF-κB and Nrf2/HO-1 signaling pathways in RAW 264.7 macrophages [29] and prevents H2O2 or Aβ1-42-induced apoptosis via attenuating JNK and p38 MAPK phosphorylation [44]. These results indicate that morroniside can play a role in the prevention and treatment of certain diseases through regulating NF-κB/MAPK in cells. It is unclear whether morroniside can act on BMSC via the NF-κB/MAPK signaling pathway. Therefore, we extracted BMSCs for in vitro experiments and established a cellular inflammation model by intervening BMSC with LPS and detected the expression of inflammatory factors IL6 and IL1β through ELISA, indicating that BMSCs were in the inflammatory microenvironment. We then performed ALP and alizarin red staining experiments and detected Runx2 and OCN mRNA and protein expressions. The results showed that morroniside could inhibit osteogenic differentiation of BMSCs, and the expression of related proteins in the TRAF6-mediated NF-κB/MAPK signaling pathway was detected through Western blot. The results showed that morroniside could inhibit the expression of TRAF6, p-p65/p65, p-Erk/Erk, and p-p38/p38. Furthermore, the nuclear translocation of p65 was detected using immunofluorescence. The results showed that morroniside could inhibit the nuclear translocation of p65.

## 4. Materials and Methods

### 4.1. Animals

Fifteen female 6–8-week-old SPF C57BL/6J mice, weighing 18 ± 2 g, were provided by Qinglongshan Animal Breeding (Nanjing, China) (Animal qualification certificate no. SCXK (Zhejiang) 2019-0002). The animals were housed at the Experimental Animal Centre of Nanjing University of Traditional Chinese Medicine. The experiment was performed after adaptive feeding for one week. The SPF conditions included routine feeding, a 12/12 h alternating light/dark cycle, constant temperature and humidity, and feed and drinking water provided ad libitum. The experimental protocol for this study was reviewed and approved by the Experimental Animal Ethics Committee of Nanjing University of Traditional Chinese Medicine (approval no. 202107A006). Twenty mice were randomly divided into four groups: control, model, morroniside, alendronate sodium groups. Except for the control group (*n* = 5), 20 female C57BL/6J mice were intraperitoneally injected with LPS (5 mg/kg) once per week for three consecutive weeks to establish an inflammatory bone loss mouse model [45]. Morroniside was administered by oral gavage (10 mg/kg body weight) once daily, whereas the control and model groups were administered the same volume of normal saline. Alendronate sodium group was given intragastric administration once a week (10 mg/kg body weight). The drug was continuously administered for three weeks, and samples were collected.

### 4.2. Micro-CT

The left femur of each mouse was placed in a cylindrical plastic specimen holder with a diameter of 20.5 mm, and the long axis of the femur was perpendicular to the X-ray source. The scanning parameters were set to a voltage of 55 kVp, a current of 70 mA, and a resolution of 9 µm. The region of interest was defined as a proximal femoral growth plate measurement of 0.3–0.6 mm, to include all secondary trabecular bone. After scanning and reconstruction, a visual 3D model was constructed, and the relevant 3D bone morphometric parameters were further analyzed: 3D bone marrow density (BMD), bone volume fraction (BV/TV), trabecular thickness (Tb.Th), and trabecular separation degree (Tb.Sp).

### 4.3. Haematoxylin and Eosin (H&E) Staining

The left femur tissue was fixed in 4% paraformaldehyde for 24 h and then transferred to a decalcification solution for four weeks. The sagittal plane was cut, dehydrated in a series of ethanol gradients, made transparent using xylene, and embedded in conventional paraffin. Then, 5-μm-thick sections were cut, followed by hematoxylin staining, 1% hydrochloric acid–ethanol differentiation, 1% ammonia antiblue, eosin staining, and neutral resin sealing, to observe the pathological structure of the bone tissue under an inverted microscope.

### 4.4. Biomechanical Parameter Analysis

The femur was selected, its length was measured along the long axis of the shaft with Vernier calipers, and the midpoint was marked. The specimen was fixed on a biomechanical tester with a mold, and the span was adjusted to 0.5 cm. The femur specimens were pressurized at a loading rate of 0.1 mm/s until the bone structure was destroyed, after which the pressurization was stopped. The maximum load, maximum deflection, and stiffness parameters were calculated according to load–displacement curves generated using a biomechanical tester.

### 4.5. Enzyme-Linked Immunosorbent Assay (ELISA)

Appropriate amounts of serum and BMSCs supernatant from each group were absorbed, in strict accordance with the ELISA kit instructions. The enzyme-labelled antibody working solution was successively added, followed by incubation at 37 °C for 60 min. Subsequently, the working and stop liquids of the substrate were sequentially added. The optical density (OD) was measured at 450 nm using an Osheng automatic multifunctional microboard card reader. A standard curve was drawn, and a regression equation was obtained. The expression of TNF-α (JinYiBai, Nanjing, China), IL-1β (JinYiBai), ALP (JinYiBai), and TRAP (JinYiBai) in the serum and TNF-α and IL-1β in the supernatant of the BMSCs was calculated.

### 4.6. Quantitative Real-Time PCR (qRT-PCR)

A total of 20 mg of tibial tissue was extracted from tibial tissue (20 mg) using centrifuge column extraction (Novozan, Nanjing, China) before freezing with liquid nitrogen and grinding into a powder. Then, total RNA was removed from the genome, and reverse transcription was performed. BlasTaq qPCR Master Mix (Abm, Zhenjiang, China) was used for qPCR analysis. Pairs were calculated using the 2^−ΔΔct^ method to determine the relative expression of mRNA. GAPDH mRNA was used as the reference gene. The primers used in this study are listed in Table 1.

### 4.7. Immunohistochemical Analysis

The paraffin sections were placed in a xylene solution for dewaxing, followed by gradient hydration with ethanol. The sections were infiltrated with a preheated closed-permeability solution at room temperature and protected from light. Sodium citrate solution was used for antigenic repair. The paraffin sections were dehydrated with ethanol, made transparent with xylene, sealed with a neutral resin, and observed under a microscope.

### 4.8. Western Blotting

The BMSCs and tibia were lysed with RIPA lysis buffer containing 1% protease, a phosphatase inhibitor, and 1% phenylmethylsulfonyl fluoride (PMSF), which was followed by protein quantification using a BCA Protein Quantification Kit (Thermo Fisher Scientific, Waltham, MA, USA). Then, 25 μg of total protein for each sample was separated using 4–20% SDS-PAGE, transferred to polyvinylidene fluoride (PVDF) membrane, blocked with 5% skim milk powder, and incubated with the primary antibody at 4 °C overnight. The following primary antibodies were used: anti-osteocalcin (1:1000) (CST; Danvers, MA, USA), anti-Runx2 (1:1000) (CST), anti-TRAF6 (1:1000) (sc-8409; Santa Cruz Biotechnology), anti-ERK (1:1000) (CST) anti-p-ERK (1:1000) (CST), anti-p38 (1:1000) (CST), anti-p-p38 (1:1000) (CST), anti-p65 (1:1000) (CST), anti-p-p65 (1:1000) (CST), and anti-GAPDH (1:10,000) (CST). The proteins were incubated with horseradish peroxidase-conjugated goat anti-mouse IgG and horseradish peroxidase-conjugated goat anti-rabbit IgG (1:10,000) (Proteintech, Wuhan, China) for 2 h and visualized using an ECL chemiluminescence reagent (Thermo Fisher Scientific).

### 4.9. Isolation and Culture of BMSCs

The BMSCs were isolated and cultured as previously described. Briefly, 10 C57BL/6J (6–8 weeks old) female mice were euthanized, and their bone marrow solution was purified using an adherence screening method. An incubation medium containing complete nutrients was used for the BMSCs (containing 10% FBS (Hyclone, Logan, UT, USA), 1% penicillin–streptomycin (Gibco, Gaithersburg, MD, USA), and α-MEM (Hyclone)). The BMSCs (10^6^/mL) were then placed in plastic culture flasks. Cells from the third passage were used for the subsequent experiments.

### 4.10. BMSC Identification

BMSCs were lightly digested and centrifuged at 3000× *g* for 5 min, suspended in a flowing buffer, and adjusted to 10^6^/mL. Fluorescent antibodies against CD90 (MBL, Beijing, China), CD29 (MBL, Beijing, China), and CD45 (Thermo Fisher Scientific), and the corresponding homologous control antibodies were added, incubated on ice for 30 min, centrifuged, washed once with phosphate-buffered saline (PBS), and transferred to a flow cytometer (BD Bioscience, Shanghai, China) for detection.

### 4.11. Cell Viability

A purity of ≥98.0% was detected for morroniside (Yuanye, Shanghai, China) using HPLC. The BMSCs were incubated with various concentrations of morroniside (0, 2, 4, 8, 16, 32, 64, 128, 256, and 512 μM) in 96-well plates at a density of 5 × 10^3^ cells per well for 48 h before evaluating using a cell count kit (CCK-8; Wako, DOJINDO, Kyushu Island, Japan). To this end, 10 μL of CCK8 solution was administered to each well, followed by the incubation of the plates for an additional 2 h. Microplate readers were used to measure the OD of each well at 450 nm (Bio-Rad, Hercules, CA, USA). Blank samples were prepared from the culture media. The cell viability was calculated as follows: cell viability = [OD (with morroniside) × OD (blank)]/[OD (without morroniside) × OD (blank)]. The highest concentration (512 μM) was found to reduce cell viability. Two concentrations of morroniside (16 and 256 μM) were thus selected for the following experiments.

### 4.12. Osteogenic Differentiation

Third-generation BMSCs in an appropriate growth state were adjusted to 2 × 10^4^ cells/well (6-well plate) with complete medium, and the culture medium was replaced with osteogenic induction medium (50 μg/mL vitamin C; Sigma-Aldrich, St. Louis, MO, USA), 5 mM β-sodium glycerophosphorin (Sigma-Aldrich), and 0.1 μmol/L dexamethasone (Sigma-Aldrich). Therefore, vitamin C needed to be administered immediately. Half the volume of the liquid was changed every alternate day. Approximately 14 days after induction, the cell morphology was observed under a microscope, and ALP staining was performed using an ALP staining kit (Beyotime Biotechnology, Shanghai, China). The induction was continued for 21 days, and alizarin red staining was performed using an alizarin red staining kit (Beyotime Biotechnology, Shanghai, China).

### 4.13. ALP Staining

After absorbing and discarding the culture medium, the BMSCs were washed twice with PBS, followed by the addition of 4% paraformaldehyde solution, fixing for 10 min, and washing three times with distilled water. A BCIP/NBT staining solution prepared in advance (3 mL of alkaline phosphatase color buffer, 10 μL of BCIP solution, and 20 μL of NBT solution) was added and incubated for 30 min in the dark. The reaction was stopped by washing twice with distilled water, and the cells were observed under a microscope and photographed.

### 4.14. Alizarin Red Staining

According to the above osteogenic induction method, differentiation continued for 21 days, and the culture was terminated when more dark yellow opaque nodules appeared in the cell culture. The medium was discarded, and the BMSCs were washed three times with PBS, fixed with a fixative solution for 20 min, washed three times with PBS, and treated with alizarin red dye solution to evenly cover the cells. The cells were stained for 30 min at room temperature, washed thoroughly with distilled water, and observed and photographed under a microscope.

### 4.15. Immunofluorescence Staining

BMSCs were induced with lipopolysaccharide (LPS) and treated with different concentrations of morroniside in 48-well plates. After 24 h, BMSCs were fixed in 4% paraformaldehyde for 10 min at room temperature and permeabilized with 0.2% Triton X-100 for 15 min at room temperature. After washing with BSA and sealing with 5% BSA, BMSCs were incubated with primary mouse anti-NF-κB p65 monoclonal antibody (Santa Cruz Biotech, Santa Cruz, CA, USA) at room temperature for 30 min. Immunoreactivity was assessed using Alexa594-conjugated streptavidin, and BMSCs were counterstained with 10 mg/mL DAPI. Cells were examined under a Nikon fluorescence microscope (Image Systems, Columbia, MD, USA).

### 4.16. Statistical Analysis

All results are presented as the mean ± standard deviation (SD) from a minimum of three replicates. Differences between groups were evaluated using one-way analysis of variance (ANOVA), followed by a Bonferroni test. Differences were considered statistically significant at *p* < 0.05. All statistical analyses were performed using GraphPad Prism version 9 statistical software.

## 5. Conclusions

This study demonstrated that morroniside inhibits bone loss in mice by activating the TRAF6-mediated NF-κB/MAPK signaling pathway, alleviating the occurrence of the inflammatory response, and alleviating LPS-induced osteogenic differentiation dysfunction of BMSCs. These data suggest that morroniside can be used effectively to treat inflammatory bone loss. This study aimed to investigate the pathogenesis of osteoporosis from the perspective of reducing inflammation, to aid in developing drugs for osteoporosis. In addition, it will contribute to the development of novel interventional targets with relevance for chronic inflammatory diseases. However, this study had some limitations. The specific mechanism through which morroniside activates TRAF6-mediated NF-κB/MAPK signaling has not been fully elucidated. Furthermore, we blocked TRAF6 via lentiviral transfection to observe whether the protein expression of the NF-κB/MAPK signaling pathway was inhibited, verifying that morroniside plays a role in promoting osteogenic differentiation through the TRAF6-mediated NF-κB/MAPK signaling pathway. This study only provides a certain reference for morroniside to slow the development of osteoporosis from the aspect of inflammation. In addition, although morroniside is a natural compound, there have been few clinical studies on it, and its absorption, distribution, metabolism and excretion in the body are still unclear, so it will take a long time for morroniside to be applied clinically.

## Figures and Tables

**Figure 1 pharmaceuticals-16-01438-f001:**
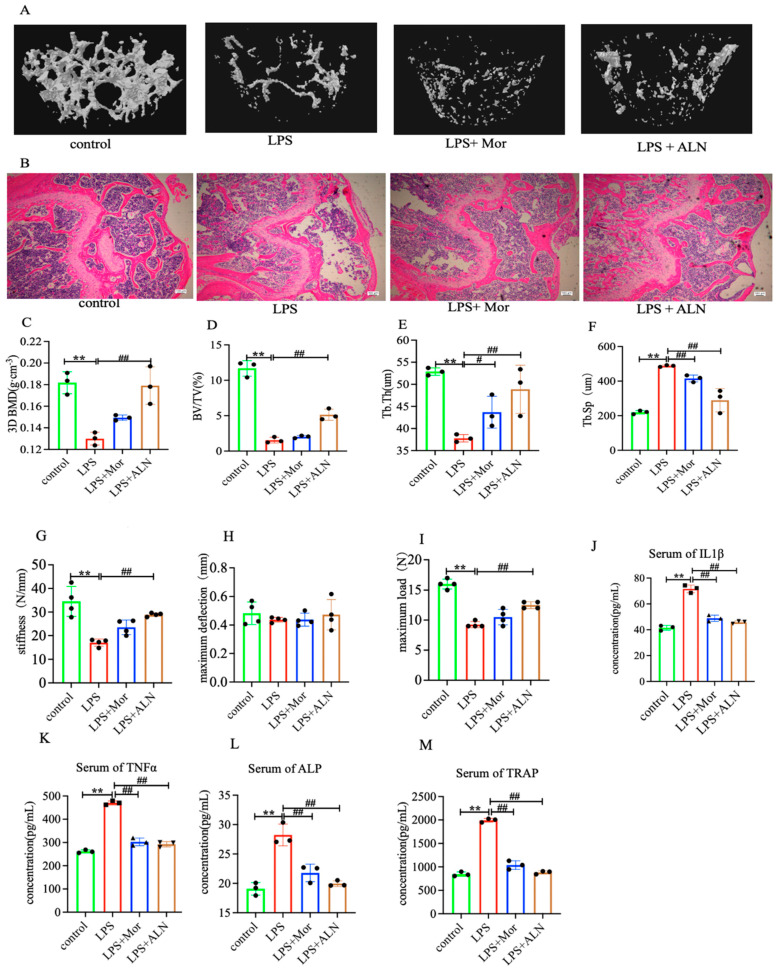
(**A**) Representative micro-CT images of left femurs. (**B**) Representative HE staining images. Scale bar = 100 μm. (**C**−**F**) Three-dimensional BMD, BV/TV, Tb.Th, and Tb.Sp. (**G**−**I**) Biomechanical test results for stiffness, maximum deflection, and maximum load. (**J**−**M**) IL-1β, TNF-α, ALP, and TRAP levels in mice sera. Data are represented as the mean ± SD of three experiments. ** *p* < 0.01 versus control group; ^#^
*p* < 0.05, ^##^
*p* < 0.001 versus LPS group. Abbreviations: ALN, sodium alendronate.

**Figure 2 pharmaceuticals-16-01438-f002:**
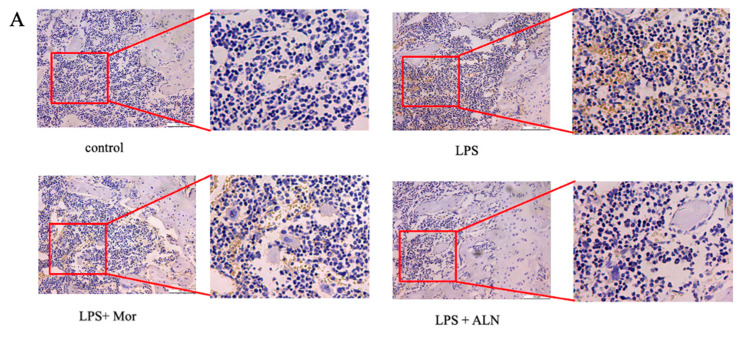
Morroniside inhibits the osteogenic differentiation of BMSCs by inhibiting the TRAF6-mediated NF-κB/MAPK signaling pathway in vivo. (**A**) TRAF6 expression was detected using immunohistochemistry. (**B**) Positive-expression area of TRAF6 in femur tissue. (**C**−**E**) TRAF6, OCN, and Runx2 mRNA expression levels in tibia tissue. (**F**) The expression of TRAF6-mediated expression of NF-κB/MAPK signaling-pathway-related proteins (p-p65, TRAF6, p-ERK, and p-p38) were detected using Western blotting assays. (**G**−**I**) The levels of TRAF6, p-p65, and p-p38 were quantified using Image J and normalized to GAPDH. (**J**) The expression of the osteogenic differentiation proteins Runx2 and OCN was detected using Western blotting assays. (**K**−**M**) The levels of p-Erk, Runx2, and OCN were quantified using Image J and normalized to GAPDH. Data are presented as the mean ± SD of three experiments. ** *p* < 0.01 versus control group; ^#^
*p* < 0.05, ^##^
*p* < 0.001 versus LPS group. Abbreviations: ALN, sodium alendronate.

**Figure 3 pharmaceuticals-16-01438-f003:**
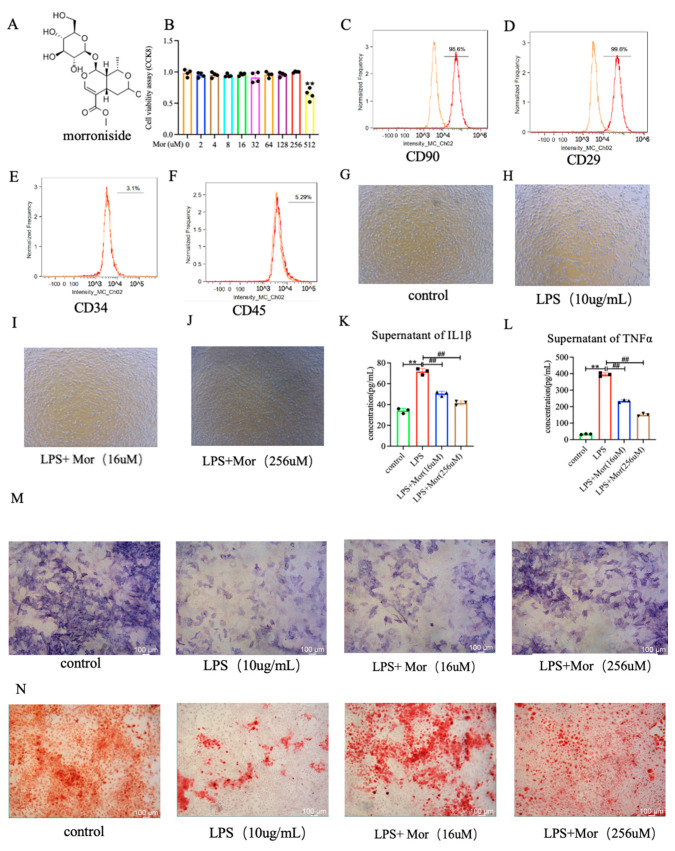
Morroniside promotes the proliferation and osteogenic differentiation of BMSCs and delays the occurrence of inflammation. (**A**) Chemical structure of morroniside. (**B**) The proliferation of BMSCs was determined at 48 h using CCK8 assays. (**C**–**F**) Identification of BMSCs with CD90, CD29, CD34, and CD45 using flow cytometry (FCM) (**G**–**J**) Cell morphology was observed under an inverted microscope. (**K,L**)The levels of IL-1β and TNF-α in the supernatant of BMSCs were determined using ELISA. (**M**) Osteogenesis was determined using ALP staining. (**N**) Alizarin red staining results. Data are presented as mean ± SD of three experiments. ** *p* < 0.01 versus control group; ^##^
*p* < 0.01 versus LPS group.

**Figure 4 pharmaceuticals-16-01438-f004:**
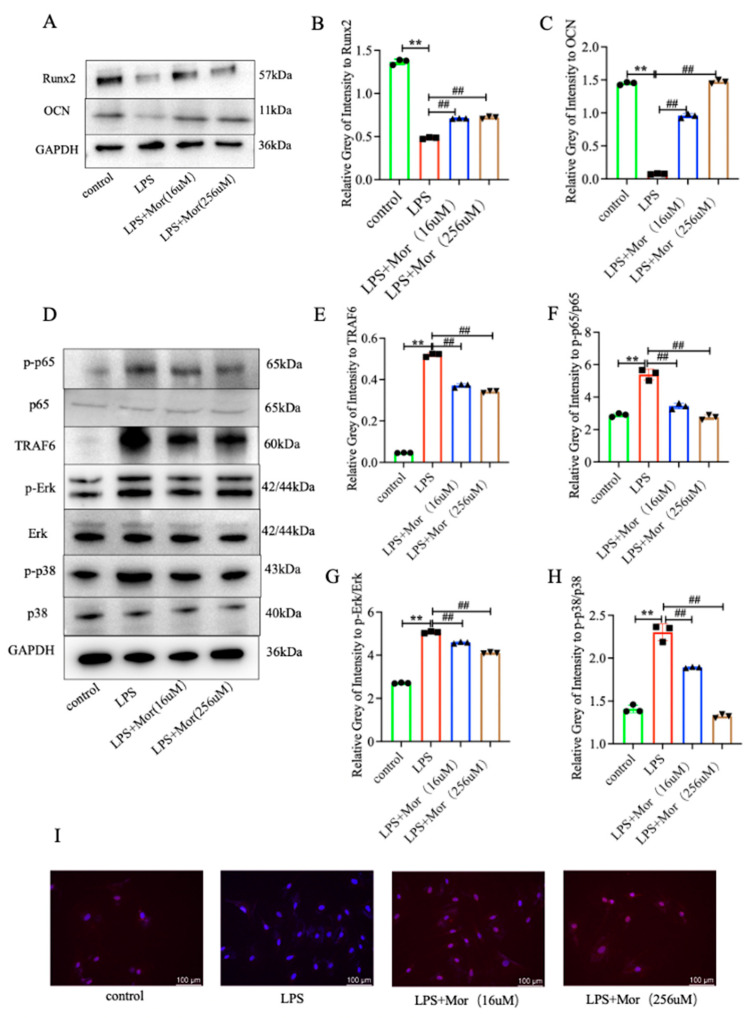
Morroniside inhibits the osteogenic differentiation of BMSCs by inhibiting the TRAF6-mediated NF-κB/MAPK signaling pathway in vitro. (**A**) The expression of osteogenic differentiation proteins Runx2 and OCN was detected using Western blotting. (**B**,**C**) The expression of Runx2 and OCN was quantified using Image J (https://imagej.nih.gov/ij/) and normalized to GAPDH. (**D**) The expression of TRAF-mediated NF-κB/MAPK signaling-pathway-related proteins (p-p65, p65, TRAF6, p-Erk, Erk, p-p38, and p38) was detected using Western blotting assays. (**E**−**H**) TRAF6, p-p65/p65, p-ERK/ERK, and p-p38/p38 were quantified using Image J and normalized to GAPDH. (**I**) immunofluorescence staining of NF-κB p65 in BMSCs, with counterstaining using DAPI, as indicated (100 μm). Data are presented as mean ± SD of three experiments. ** *p* < 0.01 versus control group; ^##^
*p* < 0.01 versus LPS group.

**Table 1 pharmaceuticals-16-01438-t001:** Nucleotide sequences of primers.

Gene	Sequence (5′ to 3′)	Application
*GAPDH*	GCCTCGTCTCATAGACAAGATG	qPCR
	CAGTAGACTCCACGACATAC	
*TRAF6*	AAAGCGAGAGATTCTTTCCCTG	qPCR
	ACTGGGGACAATTCACTAGAGC	
*OCN*	CTGAAAAGCCCACAGATACCAG	qPCR
	TGGAGAGGGTTGTTAGTGTGTC	
*Runx2*	ATGCTTATTCGCCTCACAAA	qPCR
	GCACTCACTGACTCGGTTGG	

## Data Availability

Data is contained within the article.

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
