# Peer review of "Morroniside Inhibits Inflammatory Bone Loss through the TRAF6-Mediated NF-κB/MAPK Signalling Pathway"

_pharmaceuticals, 2023, doi:10.3390/ph16101438_

Round 1

Reviewer 1 Report

The manuscript "Morroniside inhibits inflammatory bone loss through TRAF6-mediated NF-κB/MAPK signalling pathway" is interesting but some points required more work 

The abstract should be rephrased to give some numerical results 

the intro should be focused on the compound of interest and its biological activity to get inside its effectiveness 

The resolution of some figures and photos are very low to be checked 

The discussion should be enhanced by explanation of the activity based on certain functional group 

The manuscript should be checked by an English native speaker to remove some syntax and typos

Reviewer 2 Report

General comments

- Define abbreviations first time when are used in the abstract and the body of the manuscript.

- Do not use abbreviations in the titles and subtitles.

- Do not start a sentence with an abbreviation.

- The authors misuse the expression "inflammatory bone loss"

Abstract

- It is unclear which bone disease you refer to.

- The abstract is a story. The presented information did not respect the scientific rules. The aim of the study is unclear, the methodology is not appropriately presented, the results lack statistical clarity, and the presented results do not support the conclusions.

Introduction

- It is unclear the relationbetween osteoporosis and inflammation.

- "Bisphosphonates and dezumumab are currently used to treat osteoporosis" which is the effectiveness and efficacity?

- "their side effects are severe" list them as well as the associated occurrence.

- "Many natural medicines have the potential to be used to treat osteoporosis." uninformative sentence.

- "The inflammatory mediators of osteogenic differentiation in BMSCs have rarely been reported to be affected by morroniside" reference(s) needed.

- "Additionally, the specific antiosteoporotic effects of morroniside on
lipopolysaccharide (LPS)-induced osteogenic differentiation dysfunction and inflammatory bone loss in mouse models of BMSCs have not yet been reported" reference(s) needed.

- BMSCs from which species?

- "Morroniside inhibited the proliferation of BMSCs and promoted the
differentiation of osteoblasts at a certain concentration, possibly by inhibiting the LPS-induced TRAF6-mediated activation of the NF-κB/MAPK signalling pathway." this is a result.

- End this section with the aim of the study.

Results

- Figures 1 to 4. Use symbols to identify pathological changes. Describe these changes in the figure legend. Instead of column charts, please use box and whiskers. Please closely check the significance in the graphs (for BV/TV(%) show clear difference of LPS+Mor vs. control, etc.).

- "The expression of TRAF6 was additionally detected using immunohistochemistry." this information belongs to the Methods.

- "The surface markers of the third-generation BMSCs were detected using flow cytometry, and the results showed that the positivity rates of CD90, CD29, CD34, and CD45" this information belongs to the Methods section. Similar comment for "To study the cytotoxicity of morroniside, BMSCs were cultured with different concentrations of morroniside for 48 h."

Discussion

- "In this study, we established a model of LPS-induced inflammatory bone loss of BMSCs to determine whether morroniside promotes the osteogenic differentiation of BMSCs." this is a summary and duplicates information.

- "Morroniside treatment inhibited" statistically significant?

- I was not able to identify any reference to your own results.

- Explore possible mechanisms or explanations for your findings.

- Emphasize the new and important aspects of your study.

- Put your findings in the context of the totality of the relevant evidence.

- State the limitations of your study.

- Explore the implications of your findings for future research and for clinical practice .

- Discuss the influence or possible association of variables on your findings. Do not repeat in detail data or other information given in other parts of the manuscript, such as in the Introduction or the Results section.

Methods

- Please provide the approval date (ethics committee).

- Morroniside: unclear the products, when were administrated relative to the induction, how the dose was established etc.

- 10 mg/kg should be read as 10 mg/kg body weight.

- Unclear which samples were collected and when.

- Unclear what happened to the animals at the end of the experiment.

- It is unclear how the induction was verified.

Reviewer 3 Report

The paper is perfectly written the only problem is that the abbreviation  ALN used in figures is not explained in the text.

Reviewer 4 Report

I have gone through the manuscript: Morroniside inhibits inflammatory bone loss through TRAF6- 1mediated NF-κB/MAPK signalling pathway. Following are my comments:

1.     The Introduction section does not adequately explain certain elements of osteoporotic changes, such as age-related or postmenstrual osteoporotic changes.

2.     The limitations of conventional approach and possible opportunities for treating osteoporosis is very briefly explained.

3.     The findings from related studies are not discussed in introduction/discussion section.

4.     The introduction necessitates a more extended, fresh talking of recent findings and well-defined aims in a properly organised manner.

5.     Why were only female mice used in the study? Osteoporosis affects both men and women equally.

6.     The conclusion, limitations of the study or possible future directions must write under separate heading.

7.     Improve English literature of main manuscript especially of discussion.

Minor typo and language errors should be improved throughout the manuscript 

Round 2

Reviewer 2 Report

The authors followed my previous suggestion but still some changes are needed before publication:

- Please write the aim of the study at the end of Introduction.

- The column charts are not appropriate because because did not capture correctly the variability in the data. As previous requested must be changed with box and whiskers graphs.

- On histology figure, please use symbols to show the reader the pathological changes.

- Start the discussion section with your main results not with the state of the art.

- "Consultion" is misspeled.

- Refer your figures in the Discussion section.

- The limitation of the study are still not appropriately discussed.

- It is still unclear if the same reseracher addministrated all medication and performed the analyses.
